# The Role of Virtual Reality in the Management of Football Injuries

**DOI:** 10.3390/medicina60061000

**Published:** 2024-06-18

**Authors:** Andrea Demeco, Antonello Salerno, Marco Gusai, Beatrice Vignali, Vera Gramigna, Arrigo Palumbo, Andrea Corradi, Goda Camille Mickeviciute, Cosimo Costantino

**Affiliations:** 1Department of Medicine and Surgery, University of Parma, 43126 Parma, Italy; antonello.salerno@unipr.it (A.S.); marco.gusai@unipr.it (M.G.); beatrice.vignali@unipr.it (B.V.); andrea.corradi4@studenti.unipr.it (A.C.); 2Department of Medical and Surgical Sciences, University of Catanzaro “Magna Graecia”, 88100 Catanzaro, Italy; gramigna@unicz.it (V.G.); info@arrigopalumbo.com (A.P.); 3Center of Rehabilitation, Physical and Sport Medicine, Vilnius University Hospital Santaros Klinikos, LT-08661 Vilnius, Lithuania; camille.goda@gmail.com

**Keywords:** rehabilitation, sport, athletes, Soccer

## Abstract

Injuries represent a serious concern for football players, with a significant loss in terms of sport participation and long periods of rehabilitation. According to the 2019/20 UEFA Élite Club Injury Report, the average incidence of injuries during training is 2.8 per 1000 h of training, with an average absence from training of 20 days. In addition, injured athletes are 4 to 7 times more likely to relapse than uninjured athletes. High workloads and reduced recovery periods represent two of the most important modifiable risk factors. In this context, prevention and an adequate rehabilitation protocol are vital in managing injuries, reducing their incidence, and improving the return to competition. In recent years, technological development has provided new tools in rehabilitation, and Virtual reality (VR) has shown interesting results in treating neurologic and orthopedic pathologies. Virtual Reality (VR) technology finds application in the sports industry as a tool to examine athletes’ technical movements. The primary objective is to detect the biomechanical risk factors associated with anterior cruciate ligament injury. Additionally, VR can be used to train athletes in field-specific techniques and create safe and controlled therapeutic environments for post-injury recovery. Moreover, VR offers a customizable approach to treatment based on individual player data. It can be employed for both prevention and rehabilitation, tailoring the rehabilitation and training protocols according to the athletes’ specific needs.

## 1. Introduction

Football (also known as Soccer) is the most known and widely played sport in the world. According to the “*Big Count*” study conducted by *FIFA* in 2006, an estimated 265 million people play football worldwide, of which 38 million are registered with different organizations, including referees and officials [1]. Football is a complex contact sport with a relatively high risk of injury among professional athletes, amateurs, and young athletes. The evolution of modern football has led to even faster and more aggressive dynamics than in the past, requiring athletes to expend high amounts of energy and have excellent physical fitness [2].

### 1.1. Epidemiology of Injuries

A study conducted by UEFA between 2001 and 2008 on 23 professional teams participating in UEFA competitions reported a total of 4483 injuries, with 2546 (57%) occurring during matches and 1937 (43%) during training. On average, a player sustained 2.0 injuries per season. An elite team of 25 players can expect about 50 injuries each season, half of them of a minor grade and causing absences of less than a week, but as many as eight or nine being severe, causing absences of more than four weeks [3].

Muscle strains, ligament sprains and contusions were the most common, with thigh strain representing 17% of all injuries, and posterior thigh strains more common than anterior ones. Furthermore, adductor pain/strain, ankle sprain and medial collateral ligament (MCL) injuries are relatively frequent [3].

Muscle injuries constitute 31% of all injuries and cause 27% of total injury absences. In total, 92% of all muscle injuries affect the four major lower limb muscle groups: hamstrings (37%), adductors (23%), quadriceps (19%), and calf muscles (13%). Moreover, quadriceps strains (60%) affect mostly the dominant leg (preferred kicking leg) and the incidence is six times higher during a match compared with training [3,4]. Muscle injuries have been classified into non-structural and structural injuries [5]: non-structural lesions, which comprise 70% of muscle injuries in football, do not lead to theanatomical interruption of muscle fibers. In these cases, the athlete usually reports tenderness, heaviness, or a feeling of stiffness in the affected muscle, which increases with the continuation of sports activity and sometimes occurs at rest. Upon palpatory examination, stiffer muscle bundles can be found within the interested muscle. Moreover, re-injury represents a serious concern by causing more prolonged absences, and players with a history of previous injuries tend to have a re-injury risk that is 4 to 7 times greater [3].

What predisposes the athlete to a relapse is both risk factors related to the changes that occurred after the first injury (muscle hypotrophy, scar tissue, neuromuscular changes), as well as inadequate treatments (aggressive or incomplete rehabilitation, underestimation of primary injury, prolonged immobilization).

In addition to muscle injuries, one of the most common injuries concerns ligaments; in particular, anterior cruciate ligament (ACL) rupture has a serious impact on footballers’ careers. The ACL provides stability to the knee joint, preventing the anterior displacement of the tibia relative to the femur. A complete rupture is a cause of knee instability and can lead to long-term knee conditions, such as meniscosis, osteoarthritis, and a high rate of new lesions in the graft or contralateral knee [6,7]. A study based on 78 teams participating in UEFA competitions between 2001 and 2015 recorded 157 cruciate ligament injuries, of which 140 were total and 17 were partial ruptures, including six relapses and two contralateral injuries; the average frequency of cruciate ligament injuries is 0.066 per 1000 h. The full injury had a significantly higher average frequency than partial rupture (0.059 vs. 0.007 per 1000 h), and the frequency of injuries during matches was greater than 20 times compared to workout [8]. ACL injuries usually occur through traumatic mechanisms without contact, mainly during high-velocity movements, such as a quick change of direction, a sudden stop, or a landing after a jump [8].

One of the most discussed risk factors for injuries is fatigue, defined not as energy exhaustion during a game, but an effect of fatigue accumulation over a period of time, presumably due to a dense schedule of matches [9]. In addition, several intrinsic and extrinsic factors pose a higher risk of injuries, such as intrinsic biomechanical factors, hormonal, anatomical and neuromuscular risk factors, as well as limited hip and ankle mobility, which could predispose the knee to higher loads [10]. There are also some gender-related risk factors, which show that female populations have a particular predisposition (e.g., for ACL injuries), presumably due to the role of estrogen receptors and progesterone in female populations [11].

Injuries are therefore a serious problem not only for players but also for clubs as, on the one hand, the chances of the whole team performing optimally are limited; on the other hand, they also represent a cost to society (the average cost of a professional player injured for a month is about EUR 500.000) [8].

### 1.2. Injury Prevention

Injury prevention is essential for reducing the long-term consequences and minimizing the impact on the sports season. It is recommended that an approach be based on the individual athlete’s lifestyle, not solely on managing risk factors. Several injury prevention programs like FIFA 11+ and PEP have been developed in recent years [12,13].

The focus is the perception of the fatigue of athletes, which can be avoided by maintaining proper hydration, nutrition, and following individualized training protocols with appropriate loads. Moreover, muscle weakness and the difference in strength between agonist and antagonist muscles increase the risk of injury by 4 to 5 times. Therefore, it is crucial to objectively and quantitatively evaluate athletes periodically [14,15].

Prevention exercise programs with a comprehensive approach play a key role in reducing the incidence of injury. It has been shown that specific prevention programs such as Sportsmetrics and the Prevent Injury and Enhance Performance Program reduce the risk of injury and positively impact performances [16].

Eccentric exercise training is the method most used for preventive purposes because it causes a muscle adaptation that significantly improves the stiffness of the muscle–tendon unit. Moreover, proprioceptive training has also shown interesting results; it must be preceded by adequate heating, in monopodalic and bipodalic alternating support, on stable and unstable surfaces with dynamic exercises and adequate technical material [17]. Finally, selective core muscle training can also improve the neuromuscular control of the lower limbs, and working on balance has been proven effective in reducing the rate of anterior cruciate ligament injuries [10].

### 1.3. Rehabilitation

In case of injury, rehabilitation is of vital importance, with specific treatments based on the type and site of injury, and the grade of lesion. In most injuries, the treatment is conservative, with reduced workloads, active recovery with hydrokinesitherapy and the application of physical therapies [7]. The type of rehabilitation and therapeutic proposals must respect the principle of specificity, progression, and individualization. It is essential to consider pain as an indicative factor for the load capacity, with a progressive load needed to allow complete and optimal tissue regeneration [18,19]. When returning to play, it is important to apply the correct progression of sport-specific training, which is enabled by the sanitary staff’s constant monitoring and correct communication with the technical staff [6].

In case of surgical treatment, pre-operative rehabilitation is fundamental, with the aim to control pain and inflammation through cryotherapy, compression and anti-inflammatory drugs, the recovery of ROM through active and passive mobilizations, and the strengthening of quadriceps and hamstrings through exercises in the closed kinetic chain, possibly by using electrical stimulation and proprioceptive education [20].

### 1.4. Virtual Reality

Alongside the classic rehabilitation techniques and various protocols for injury management, we have observed the swift advancement of new technologies such as virtual reality (VR). VR can be defined as “a medium comprising interactive computer simulations that detect the position and movements of the subject, giving the feeling of being mentally immersed in the virtual world”.

Through various tools, including headsets, gloves, shoes and joypads, the experience is enriched, and the three-dimensional environment appears real [21].

It is possible to distinguish three types of virtual reality, each with its specific characteristics [21,22]:Immersive VR creates a three-dimensional room in which the subject is immersed with no reference to the real world.Non-immersive VR is generated through computers and consoles, allowing the visualization of 2D virtual objects in the real context.Mixed reality combines immersive and non-immersive virtual reality: the technology is generated by head-mounted displays that overlap with the real-world 3D virtual objects (holograms), allowing interaction with them.

In the medical field, VR has shown interesting results in numerous applications, mainly in motor and cognitive rehabilitation and the therapy of psychiatric disorders, such as anxiety and depression [23]. Currently, VR-based rehabilitation is used particularly in the neurological field for balance disorders and impairment of the upper limbs following stroke [22].

This technology has become increasingly popular in recent years, as it provides a secure and controlled environment for performing various actions. In addition, using VR constitutes a motivating and engaging rehabilitation approach for the patient, with the possibility to customize the treatment based on the patient’s needs [9].

Among the critical issues that emerge from the use of VR in healthcare are the costs of equipment, the need for technical expertise, and cybersickness, defined as the nausea and headaches caused by the prolonged use of smartphones, tablets and headsets [24].

In sport, VR can be seen as an assistive technology that provides specific support for training through the simulation of game dynamics and data analysis. However, the applications of VR vary, and include the improvement of teamwork and reductions in the risk of injury throughout a personalized training program for a single player based on the data collected, helping them to improve their weaknesses and reach their potential [24,25].

In addition, athletes who undergo a rehabilitation program involving VR describe the experience as rewarding, pleasant and motivating, increasing their compliance [26].

VR’s interactive and immersive nature can aid in conducting training sessions and enable accurate statistical analysis. Athletes can practice on a simulated field using headsets and head-mounted displays while collecting and analyzing crucial data in real time, e.g., heart rate and blood pressure, to evaluate the effectiveness of their training [24].

To the best of our knowledge, no previous study has summarized the role of VR in the management of football injuries. Thus, the aim of this narrative review is to provide a broad overview of the currently available VR approaches used to reduce the incidence of injuries and improve the rehabilitation outcome, promoting a personalized strategy for the multidisciplinary management of footballers. In details, this review analyzes the role of VR for preventive purposes through a study of the biomechanics of athletic gestures in a realistic environment and the reduction in concussions; the rehabilitative purposes, improvements in pain symptoms, return to sport, and kinesiophobia are also analyzed.

## 2. Materials and Methods

### Search Strategy

This narrative review involved three electronic databases, namely PubMed, Scopus and Web of Science, from inception to 10 February, following the search strategy shown in Table 1. Moreover, additional articles were selected from the bibliography of the included studies.

We selected articles considering footballers as the population (P); considering the use of VR as intervention (I); and considering injury management, biomechanics and concussion as outcomes (O). We included articles with the full text available and written in English.

Two independent reviewers screened articles according to their title and abstract. The full text of the articles selected was reviewed in accordance with the selection criteria. Two reviewers extracted the data from the selected studies on a Microsoft Excel sheet. In case of disagreement, consensus was reached with a third reviewer.

## 3. VR in Injury Prevention

VR has immense potential in the field of preventive medicine, as it allows for the creation of personalized, immersive exercises that can effectively prevent injury. By simulating various scenarios that focus on specific athletic gestures, training programs become more engaging, safer, and better tailored to the sport. Additionally, VR serves as a valuable research tool for assessing risk factors in conditions that closely resemble those experienced in the field. By recreating virtual environments and protected game scenarios that mirror real-life situations, it is possible to collect and analyze biomechanical and kinematic data close to the match dynamics. See Table 2 for further details.

### 3.1. Cognitive Ability and Speed of Execution

Football is constantly evolving, athleticism is increasingly important, and the speed of the game is much faster than in the past; this leads to increased cognitive engagement, and quick and accurate decisions must be made based on retrieving and processing information from a dynamic environment. Therefore, football players must improve their cognitive–perceptual skills to avoid the adverse effects of mental fatigue during training and matches. Superior perceptual–cognitive motor skills promote the formation and development of motor skills, and high levels of perceptual–cognitive spatiotemporal ability can effectively improve performance [27,28]. In football, the player must move the neck, head, and eyes to perceive the position of teammates and opponents in the environment and then make the most appropriate decision based on the current situation. The processes of anticipation and decision-making are the key skills of a football player. Namely, anticipation is the ability to recognize the outcome of other athletes’ movements before they are executed; decision-making is defined as the ability of the human brain to extract meaningful contextual information from the visual scene, and visual exploration behavior is part of the mechanisms behind decision-making. Anticipation and decision-making are important requirements for players because of the variable and high-velocity environment [29]. The game is intense, the situation on the field is fluid, and players must react quickly and accurately depending on the situation [30].

In this regard, VR could allow video stimulation training to improve the perceptual-cognitive abilities of players, creating a virtual world with multiple sensory experiences, e.g., changing the number of opponents or teammates and the game situation. In this way, it is possible to improve the choice of passage, the speed that is used, and the fixation characteristics of the most skilled footballers. Moreover, improving their scanning behavior could be especially useful for central midfielders. This would allow them to scan more, and a higher frequency is associated with a higher probability of completion [31,32]. The transferred skills resulting from VR perceptual–cognitive training could be proportional based on the degree of similarity between the training modality and the targeted skill. Moreover, the results of a study in the literature showed that VR is superior to other forms of training based on 2D video stimulation since it provides more immersive and engaging information, optimizes task representation, and effectively improves cognitive–perceptual skills [24,27]. However, despite efforts to create environments that are as realistic as possible, discrepancies with real life remain inevitable.

In performance analysis, superior performance is readily apparent on observation, while the perceptual–cognitive mechanisms that contribute to the expert advantage are less evident; therefore, the ability to create experimental tasks and conditions that allow expertise to emerge is a critical factor in players’ selection, and can be defined into specific parameters of sports performance [33]. The literature has identified the most prevalent outcome measures concerning perceptual–cognitive differences between expert and nonexpert athletes [34], including response accuracy, response time, and eye movement features. In this scenario, VR can give consistent support in order to distinguish participants based on their levels of expertise. For instance, the number of fixations, the fixation duration, and quiet eye periods analyzed by a machine learning model can be highly informative to distinguish different classes of football goalkeeper prowess [35]. Moreover, it has been found that by subjecting novice, academic and professional players to a range of virtual reality-based football drills, the VR simulator can significantly differentiate across all expertise groups through an algorithm that calculates the accuracy of the technical gesture [36]. By requesting football players to perform a specific action, such as a pass through a virtual simulator, it is possible to extract the passing accuracy (number of correct passes and the accuracy of these passes), reaction time (how long players dwelled on the ball before making a passing decision), composure (maintaining performance level despite increases in task difficulty), and adaptability (the number of touches with both feet) so that the athlete most skilled at the task can be determined. Indeed, with the possibility of categorizing players into class A, B or C based on their proficiency, it is clear that VR could play in important role in the football market. Implementing a systematic VR approach and choosing more appropriate young players in the context of talent scouting could positively affect the management of club budgets and targeted spending on quality athletes [36,37].

### 3.2. Psychological Aspect

Stress and competitive anxiety are two aspects that have received considerable attention in the world of sport. Stress occurs when there is an imbalance between the physical and mental demands placed on the athlete and the ability to respond in such conditions; specifically, psychological stress is the relationship between the person and the environment that the person perceives as straining or exceeding his/her resources and endangering their well-being [38]. One factor that contributes to psychological stress is competitive anxiety, namely the response to a sports-related stressful situation that results in a series of cognitive appraisals and behavioral responses. When it becomes too high, the athlete may become excessively excited, exceeding their mental ability to process stimuli, resulting in increased stress and decreased performance. Competitive anxiety is divided into cognitive anxiety (negative thoughts and concerns) and somatic anxiety (physiological signs of nervousness and tension) [39]. Therefore, athletes must modulate psychological stress and anxiety by developing or discovering positive coping mechanisms for optimal performance and well-being. Inadequate coping mechanisms indicate poor adaptability to stress. A negative response to stress and anxiety impairs motor coordination, reduces flexibility, increases cognitive and somatic anxiety, and shifts attention to narrow the field of vision. The literature has extensively analyzed the relationship between sports gesture anxiety; it has been found that an increase in anxiety is associated with the deterioration of sports performance. It was seen, in fact, that the precision of penalties is inversely proportional to the state of seniority; in addition, players’ speed and swing in table tennis worsened with an increase in anxiety [40,41]. Athletes’ rising psychological stress and anxiety levels have prompted the implementation of several interventions. These interventions include awareness, relaxation imagery, deep breathing and muscle relaxation [42]. With the increasing use of technology, technological devices have started to be used in the treatment of anxiety. An immersive visual experience can result in better results in reducing anxiety compared to traditional relaxation techniques. In this context, VR has already been used in both non-sport and sport contexts for the treatment of anxiety, with excellent results in terms of relaxation, allowing better coping mechanisms to be developed in the professional context [42,43,44].

Furthermore, the ability to recreate customized game scenarios, complete with distracting factors like stadium noise and opponents, can be a valuable tool in psychological training, especially in youth teams. Recent studies have shown that the presence or absence of spectators in a football stadium can significantly impact the psychological state, behavior and performance of players and football teams [31,45].

In the contemporary sports landscape, mental coaches have become integral to both individual and team sports. This is due to increasing pressure on athletes and the recognition that the mental aspect should be trained on par with the physical component. In this perspective, VR can be a valuable tool for preparing athletes for stressful environments. It can also positively influence an athlete’s perception of anxiety during high-pressure situations, such as taking a penalty kick, and help in reducing the heart rate [31,42,45].

### 3.3. Biomechanics Evaluation and Neuromuscular Training

At the forefront of sports medicine, the novel application of VR technology is revolutionizing the way we analyze in-game performance data. By immersing athletes into meticulously crafted virtual environments, researchers can simulate realistic scenarios, enabling them to analyze the athlete’s movements and identify potential risks factors. VR technology can be a valuable tool for detecting biomechanical issues specific to certain gestures that pose a risk of injury for athletes. The main advantage of this technology is its ability to provide a contextualized view of gestures within a particular sports environment. For example, in contact sports like football, numerous variables impact the athlete’s kinematics and biomechanics, such as the need to avoid or counter an opponent. VR can be helpful when athletes need to analyze the angles of their hip, knee, and ankle joints to evade opponents, such as when an opposing player is running toward them [25,46].

For example, it is possible to assess differences in the biomechanics of athletes during sport-specific activities, such as jumping and landing, actions commonly executed in a football game. In this case, the physician can analyze biomechanics in a VR laboratory highlighting, e.g., the differences between a standard vertical drop jump and a VR jump to head a ball in a football-specific corner kick scenario [25]. In a study by Di Cesare et al. [25], authors showed different values in the two conditions analyzed. In the VR scenario, they registered a reduction in hip and ankle flexion in the sagittal plane and a reduction in hip and ankle excursion at the peak knee flexion, compatible with a more stiff and possibly dangerous landing [47]. This could pave the way for a deeper understanding of game biomechanics and related risk factors.

Moreover, this technology allows athletes to rewatch and examine their movements during the session for a patient’s self-correction. The advantage of this diagnostic–preventative–monitoring tool is the ability to perform the same exercise several times under the same conditions, providing new insights [46].

Furthermore, it is possible to create an augmented neuromuscular training (aNMT) program to enhance proprioception and neuromuscular control. Namely, aNMT aims to reduce injuries utilizing biofeedback, giving the participants visual stimuli while they execute movements [48]. In the late rehabilitation phase, VR represents a tool to test the transferability of the training in the context of sports games, monitoring the athlete’s movements performed in a safe environment.

In one study by Kiefer et al. [49], it was possible to recreate a real side-cutting defensive action performed against a non-player character (NPC) within a VR laboratory in which athletes could perform exercises as if they were in a football field.

The visualized virtual environment was simulated using Unity 3d PRO, and participants viewed the environment through a custom-built wireless HMD to allow them a greater degree of freedom. In the proposed scenario, the athlete had his back to the goal in a football field. At the sound of a whistle, the defensive action began against the NPC, advancing toward the goal. The athlete had to run toward the NPC and counter his action as it quickly changed direction. This created unexpected lateral cutting actions, which varied according to the virtual player’s movements [49].

In another study by Grooms et al., the athletes underwent specialized training that relied on a visual stimulus biofeedback. The purpose of this training was to identify and address mechanisms that could potentially lead to injury [50]. Specifically, during closed-chain exercise (e.g., squat, single-leg Romanian deadlift, tuck jumps), a rectangle that represented the perfect biomechanics associated with the performance of that exercise was displayed on a screen, and each time the system detected biomechanics that pose a risk to ACL damage, the rectangle started deforming, thereby inducing self-correction in the participant. At the end of the training period, changes in lower motor cortex activity were detected at the fMRI of patients performing a leg press movement with and without a load [50].

### 3.4. Football Heading

Research has suggested a potential link between repetitive football heading and an increased risk of neurodegenerative pathologies in athletes. While further studies are required to confirm this correlation, it has been advised as a precautionary measure to reduce the frequency of heading drills during training [51].

Limitations have been established regarding the number of football heading movements allowed during training and matches. For example, the Football Association of England suggests that players over fourteen years old should not exceed ten headers during a once-a-week session. For kids up to the age of eleven, it is recommended that headers are not used at all [52].

Moreover, UEFA introduced “UEFA heading guidelines for youth players”, a series of guidelines intended to better manage heading during training by teaching the correct way to tackle heading drills. However, the most important modifiable factors are the ball size and pressure, neck strengthening, less frequent trainings and education [2]. Nevertheless, this limits the possibility of training players to perform headers, lowering the player’s ability to direct the ball correctly. Marshall et al. [53] discovered that the group of players who were trained in headers using VR experienced a significant increase in the number of goals scored with headed shots compared to those in the control group. Additionally, the VR group reported an increase in perceived confidence and self-efficacy before the second test [53].

VR technology enables the creation of simulated game environments that can be leveraged for training purposes. The use of a point-based system can facilitate the establishment of a ranking system, which can potentially foster healthy competition among participants. Thanks to VR’s ability to simulate diverse and dynamic scenarios, it is possible to populate the environment with NPCs that can engage in offensive or defensive actions, exhibiting both predictable and unanticipated behavior. Balls can be thrown, and their speed, trajectory and acceleration can be manipulated [53]. Moreover, it could be possible to integrate the lack of sensory feedback to the head with tactile stimulations for a more realistic experience [54] However, VR is a promising compromise between the concussion prevention guidelines and performance.

**Table 2 medicina-60-01000-t002:** Injury prevention.

References	Sample	Duration	Intervention Volume	VR Technology	Aims	Variables	Findings
Fortes et al. [27], 2021	26 male, 15.4 y.o.	8 weeks	18 sessions with 20 video clips (M = 10.2 s) per session (=1 h)	Utopia 360 with an LG3 smartphone	To compare the improvements in young football players with VR training to improve perceptual–cognitive skills	Passing decision-making (appropriate vs. inappropriate decisions); visual search behavior; inhibitory control performance	Both training groups significantly improved in all three variables, with the VR group showing significantly greater improvements than the VID group for passing decision-making and visual search behavior.
Hosp et al. [35], 2021	Total 35 male:12 expert goalkeepers (mean age = 16.60); 10 intermediates Goalkeepers (mean age 22.00); 13 Novices goalkeepers (28.64 y.o.)	Single session	52 trials per subject	HTC Vive, a consumer-grade VR headset	Classification of goalkeepers between three stages of expertise	Passing decision-making (appropriate vs. inappropriate decisions); eye tracking	The parameters evaluated by the analysis of eye movement made it possible to classify the goalkeepers between three stages of expertise.
Wood et al. [36], 2021	Total 51 players; 17 professional players (13 males, 4 females; 28.41 y.o.); 17 academy players (14 males, 3 females; 14.47 y.o.); 17 novice players (9 males, 8 females; 21.53 y.o.)	Single session	Not specified	The MiHiepa Sports Rezzil VR platform	Assessment of perceptual–cognitive processes needed for expertise in football; differentiated professional players, intermediate players, and novice players	Rondo scan; color combo; shoulder sum; pressure pass	VR simulator differentiated professional players compared to both academy and novice players.
Harrison et al. [42], 2021	13 females; 19–22 y.o.	Single session	3 blocks of five penalty kicks against a goalkeeper in three different condition: baseline, high stress situation, VR intervention	Liminal VR	VR relaxation effects before competition	MRF-3; RSME; DELSYS Trigno Avanti; HR; the Igroup Presence Questionnaire	Perceived anxiety levels were reduced significantly after VR relaxation intervention.
Mc Leod et al. [55], 2008	Test 1: 2 males, 19 y.o.; Test 2: test 1 + 2 males, 26 and 28 y.o.	Single session	12 blocks of 40 trials per participant	The Datavisor 80	football players heading analysis	Players’ responses to a step reduction in dα/dt when they had to move backwards or forwards but not laterally to head the ball; players’ response to a step change in dα/dt when they had to move laterally	Players intercepting balls use servo control strategies.
Cortes et al. [46], 2011	13 females, 19.3 ± 0.9 y.o.	Single session	5 successful trials	Microsoft Visual Cþþ2005 Express Edition	Assessment of the differences between unanticipated and anticipated lower extremity biomechanics while performing a sidestep cutting task in VR	Running stop; Sidestep cutting	The unanticipated sidestep cutting task had different neuromechanical characteristics than the anticipated condition, with increased knee abduction angles, knee internal rotation, and hip abduction, and decreased knee flexion angles.
Dicesare et al. [25], 2020	22 females, 16.0 ± 1.4 y.o.	Single session	4 successful trials	Custom-built for immersive VR (unspecified device)	Examination of the biomechanical injury risk during a jump–landing task in VR	Hip, knee, and ankle joint kinematic differences in the frontal and sagittal planes	Reduced hip and ankle flexion, hip abduction, and frontal plane ankle excursion
Kiefer et al. [49], 2017	5 females, 16.11 ± 1.52 y.o.	4 weeks	4 trials at T0 = baseline; 4 trials at T1 = week 8	The sport-specific virtual environment was built and displayed using Unity 3D Pro	To assess injury risk during performance of sport-specific VR tasks	Hip and knee joint kinematic differences	VR can be used to assess the risk of injury when simulating real sports performance.
Marshall et al. [53], 2023	36; CG = 16 males and 2 females, 28.67 ± 5.95 y.o.; IG = 14 males and 4 females, 24.17 ± 5.02 y.o.	10 Days	15 trials at T0; 15 trials at T1; IR = 3 training sessions between T0–T1	The Oculus Quest 2	Efficacy of training football heading in immersive VR	goals scored; shot accuracy; perceived confidence; perceived self-efficacy	VR group significantly improved heading performance. Training in VR also had significant benefits, improving perceptions of confidence in general heading ability and perceptions of self-efficacy.

Abbreviations: VR = Virtual Reality; CG = Control group; IR = Intervention group; HR = Heart Rate; RSME = Rating Scale for Mental Effort; MRF-3 = Mental Readiness Form-3; y.o. = years old.

## 4. VR in Post Injury Application

In the post-injury phase, VR has shown interesting results in tailoring exercise plans to athletes’ needs, by recreating sport-specific scenarios in a safe environment to bridge the gap between rehabilitation and the reathletization, ameliorating the emotional impact of the injury and lowering the level of kinesiophobia. See Table 3 for further details.

### 4.1. Low Back Pain

Chronic low back pain (LBP), persisting for more than 3 months, is a prevalent and significant issue in football, ranking as the fourth most common disorder with an incidence rate of 32–42% per year and with higher incidence in professional athletes [56,57]. The high workloads due to training and competitions cause mechanical stress to the musculoskeletal system. The diagnosis and treatment of chronic LBP is controversial and there is no clear consensus on its optimal management. In addition, most LBP in football players has no identifiable etiology. Several causal factors for LBP have been identified, including the instability of the trunk muscles, the incoordination of the hip spine during movement, and the repetition of excessive lumbar extension and rotation [58]. Trauma to the soft tissues of the lumbar region during athletic activity can affect the function of the central muscles and lead to muscle imbalance. Changes in central muscle function can lead to a wide range of compensatory movements, which can lead to lower back pain. Also, improper activities during the training session may be due to abnormal muscle activity, which may further aggravate the condition [57]. The movements involved in kicking the ball, placing the support foot close to the ball and rotating the trunk at high speed, cause significant stress to the lumbar spine and hips. Compared to an instep kick, a lateral kick increases the external angle of rotation of the hip, while an outstep kick increases the internal angle of rotation of the hip. During a football movement, the presence of LBP shifts the player’s center of mass (COM) laterally and rotates the lumbar spine significantly, resulting in additional stress on the lumbar spine [58,59].

In this context, VR can provide valuable support to rehabilitation programs, allowing the creation of an exercise plan that aims to improve the functional ability of players through neural facilitation, the stimulation of the sensory functions, and motor learning. For the upper body, it is possible to plan exercises that improve the core stability and balance of abdominal muscles and spinal erectors for proper muscle balance in order to prevent and/or treat sports disorders such LBP [31,57,60].

With the use of a virtual platform in an upright position or sitting position, it is possible to perform the movements of torso bending, lateral extension and flexion, selectively controlling the speed and difficulty of execution by working on the neuromuscular system to facilitate the activity of the motor unit [61]. A HDM or joystick would allow movement tracking to be performed, and the difficulty of serious games can be modified according to the degree of disability with increasing speeds or a range of motion. The effectiveness of VR in rehabilitation programs is supported by scientific evidence. In detail, following a treatment program with VR, an improvement in pain, quality of life, and jumping performance that is associated with a reduction in inflammatory biomarkers such as C-reactive protein (CRP), tumor necrosis factor (TNF-α), and interleukins has been demonstrated. In addition, MRI and ultrasound have shown an increase in the cross-sectional area of the multifidus, psoas, and erector muscles of the spine [57,62].

### 4.2. Concussion

Sport-related concussion (SRC) is a type of mild traumatic brain injury that occurs when head trauma results in transient neurological impairment, observed commonly both in professional and amateur sports [63]. The Consensus Statement from the Berlin 2016 International Conference on Concussion in Sport has provided a logical flow of clinical concussion management and considerations referring to the ‘11 Rs’ of SRC (recognize, reduce, remove, refer, re-evaluate, rest, rehabilitate, recover, return-to-learn/return-to-sport, reconsider and residual effects). A similar format has been followed for the Amsterdam 2022 Statement, with additional ‘Rs’ including RETIRE, to address issues related to potential career-ending decisions, and REFINE, to highlight the need to embrace ongoing strategies to advance the field [63].

In football, though not publicly perceived as the sport classically associated with head trauma and concussion, head injuries are mostly caused by head-to-head collisions and unanticipated ball contacts, with SRC presenting 22% of football-related injuries. Accordingly, 63% of football players have suffered a concussion during their playing careers and 82% of athletes who have suffered a concussion have experienced two or more concussions throughout their playing careers [64].

Exposure to acute concussion, repetitive concussions, as well as cumulative “sub-concussive” result in athletes having an increased risk of developing long-term neurologic impairment, post-concussion syndrome, and structural brain changes detectable on functional brain imaging [65,66,67,68]. Furthermore, having ascertained that concussed athletes in American football have increased odds of sustaining an acute lower extremity musculoskeletal injury after returning to play than their non-concussed teammates, a possible connection between subclinical head injury and impairment in proprioception has been suggested [69]. Therefore, SRC represents a problem for the health of athletes due to its neurological and orthopedic consequences. Data concerning the recognition of the symptoms of SRC suggested that only 19% of patients realize that the symptoms they complain of constitute a concussion [64,70,71]. This indicates a lack of player education regarding the topic. For this purpose, VR seems to provide a novel tool for instructing athletes to recognize the most common symptoms of SRC. Through the use of semi-immersive VR, it is possible to educate athletes on the effects of SRC, creating an ideal environment in which the player is immersed in the virtual environment as a football athlete playing on an outdoor field when a virtual football ball unexpectedly hits them in the head, causing them to start experiencing various simulated concussion symptoms (e.g., blurred vision, light sensitivity, headache, and nausea) both on the field and after virtually returning home. Filmed video clips and scenarios of actual preteen football players reinforce the virtual-to-physical relationship by allowing users to see familiar, relatable projections of themselves and their environment in the virtual space [72].

The use of VR technology to teach players, especially adolescents, about concussions offers many potential advantages over traditional approaches to concussion education. First, VR technology allows athletes to virtually feel and experience SRC in a safe, computer-generated environment. These manipulations provide an impactful, experiential, and active learning environment, allowing them to enhance their knowledge through experience [72,73].

In addition, VR can be used to assess the return to play of a football player who has undergone an SRC. The need to assess the condition of the athlete after the injury is crucial, to ensure the feasibility of returning to the field in concert with the safety and health of the athlete themselves. In this regard, the Virtual Reality Football For Concussion Assessment has been designed to measure athletes’ performance on immersive, sport-specific tasks to improve return-to-play decision-making after a concussion [74].

### 4.3. Kinesiophobia

It is crucial to consider the emotional well-being of athletes in addition to their physical recovery when formulating rehabilitation plans. The rehabilitation approach should not only focus on restoring the physiological and physical condition, but also address the psychological impact on the athletes. A crucial factor that should not be overlooked in the rehabilitation of post-injury football players is kinesiophobia. Kinesiophobia is an excessive, irrational fear of physical activity or movement due to the increased pain sensitivity caused by a painful injury or damage to the body [75]. It may seriously effect recovery, leading to increased muscle weakness, increased pain and a decreased level of daily activities [76].

Some studies have highlighted the moderate prevalence of kinesiophobia in football players [76,77]. At present, nonpharmacological therapy is the preferred approach for people with chronic pain [78]. Some intervention methods have been studied, such as cognitive behavioral therapy and graded exposure therapy. However, they reduce the level of kinesiophobia without substantially reducing the pain, which still inevitably affects the rehabilitation process, leading to low compliance and unsatisfactory recovery effects. Exercise, which is recommended as an effective treatment in the context of post-injury rehabilitation [79], may also cause pain to increase dramatically during the sessions, contributing therefore to kinesiophobia. In this scenario, VR has been shown to significantly reduce the level of kinesiophobia, providing rehabilitation training in a novel and attractive way through the creation of a virtual environment that the user can interact with, with a consequently positive impact on the emotional state of the athletes. VR provides the possibility of task-oriented and multisensory feedback training. This facilitates visual, auditory and tactile input, resulting in more accurate and complete motion feedback, exercise capacity development, and exercise compliance improvement. Furthermore, as a complementary therapy, VR has shown more improvements than exercise alone in balance and gait, reducing the fear of falling and promoting the recovery of limb function. In this regard, virtual reality-guided imagery (VRMI) is a great example of how this technology can be implemented to promote psychological readiness and facilitate a smooth transition back to sport [80]. Imagery, specifically, which involves an athlete replicating reality by mentally reviewing an action and picturing visual, auditory, tactile, olfactory, and even gustatory cues [81], promotes movement repetition and improves motor abilities in different virtual settings [82]. VRMI based on video-game-based exercises contributes to movement facilitation and functional training in ludic ways, and helps to alleviate psychological distress, including fear of re-injury and pain perception, in first-time ACL restoration patients [83], resulting in more favorable outcomes.

Nevertheless, it is important to mention that the inclusion of a high-tech VR tool in psycho-correctional programs, while it has been demonstrated to be useful for overcoming the fear of movement and pain of neuropathic and mixed genesis, remains ineffective in relieving nociceptive pain (i.e., pain where there is no psychological component); therefore, the organic component of the injury should not be neglected [62].

One further specification, in conclusion, is about the different natures of VR. Indeed, compared with fully immersive virtual reality, which would provide a better immersive experience and should theoretically achieve better results, non-immersive virtual reality has demonstrated better intervention effects in the context of kinesiophobia [84]. Fully immersive virtual reality requires the use of a head-mounted display as well as a fully occluded peripheral vision, and some participants may experience motion sickness. For this reason, non-immersive virtual reality may be a better option for rehabilitation training, considering its acceptance among the population and its lower cost [85].

**Table 3 medicina-60-01000-t003:** Post-injury application.

References	Sample	Duration	Intervention Volume	VR Technology	Aims	Variables	Findings
Nambi et al. [60], 2020	45 males; aged between 18–25	4 weeks	5 sessions of 30 min per week	ProKin system PK 252	Comparing the effect of VR training, isokinetic and conventional training on clinical (pain, wellness) and athletic performance (sprinting, jumping) of university football players with chronic low back pain	Pain intensity; Sprint performance; Jump performance, countermovement jump, squat jump	The VR group had significantly higher improvements in pain intensity, player wellness, and sprint performance than the other two groups
Nambi et al. [57], 2022	60 males; aged between 18–25	4 weeks	5 sessions of 30 min per week for 4 weeks	ProKin system PK 252	Comparing the effects of VR, conventional programs and isokinetic programs on imaging findings and inflammatory biomarkers in football players with chronic non-specific low back pain	Pain Intensity; Cross-sectional area (CSA); Muscle thickness; Inflammatory biomarkers	A significant improvement in the pain intensity and biomarker measurements for the VRE group was observed; Isokinetic group showed a greater number of significant changes in muscle CSA and muscle thickness
Nambi et al. [61], 2021	60 males; aged between 18–25	6 months	5 sessions of 30 min per week for 4 weeks	ProKin system PK 252	Determining and comparing the effects of VR and core stabilization training on physical efficiency in football players with chronic low back pain	Sprint Performance; Jump Performance; Quality of life; Pain	VR reduced the pain status by changing the inflammatory mechanism compared to the core stabilization and control groups
Nambi et al. [62], 2021	54 males; aged between 18–25	6 months	5 sessions of 30 min per week for 4 weeks	ProKin system PK 252	Determining the short-term psychological and hormonal effects of VR training on chronic low back pain	Psychological variables (pain intensity and kinesiophobia): Hormonal variables (glucose, insulin, growth hormone, prolactin, ACTH, and cortisol)	VR group showed more significant changes in pain intensity and kinesiophobia than the combined physical rehabilitation and control groups at T1 and T2
Sullivan et al. [72], 2024	33 Children; aged between 9–12 (20 male, 13 female)	single session	10 min per session	ProKin system PK 252	Developing a VR concussion education app	RoCKAS-ST; Attitudes toward concussion: reporting: 8 questions using a 5-point scale	No statistically significant improvements in attitudes toward reporting and reporting intentions following the VR session were observed

Abbreviations: VR = Virtual Reality; ACHT = Adrenocorticotropic hormone; RoCKAS-ST = Rosenbaum Concussion Knowledge and Attitudes Survey-students.

## 5. Conclusions

Despite its lack of popularity in football, VR has been a valuable tool in rehabilitation for years. The increasing development of technology has inevitably led to the progressive use of innovative tools to improve medical knowledge and positively influence athletic performance.

This narrative review highlights the role of VR in rehabilitation and research in football. Its main advantage includes the possibility of recreating the sport-specific technical gestures as closely as possible. The movement analysis of the athlete and a better understanding of the biomechanical risk factors could pave the way for an improvement in the treatment protocols used for injury prevention and rehabilitation. Moreover, thanks to the amount of data collected, it is possible to develop a personalized exercise plan based on the athlete’s needs. Additionally, VR represents a valuable tool for investigating and stimulating the cognitive and neuromuscular ability of the footballer, who has to face a high-velocity environment during matches. Indeed, VR environment has the advantages of a standard controlled environment in which the patients can perform exercises subjected to multisensorial stimuli, close to the on-field context. Through virtual reality it is also possible to recreate interacting scenarios, which can educate athletes on how to properly recognize, train and recover from potentially harmful football-related situations, lower back pain, and kinesiophobia in the post-injury phase. Moreover, VR represents one of the few methods that respects the guidelines for concussion prevention, as recommended by the English Football Association, allowing a reduction in concussion-related effects on the brain and the possibility of training players in heading. It could be suggested that the head-mounted displays are improved for a more realistic experience during headings. However, there is still a lack of standardized protocols and a need for a randomized controlled trial to evaluate the efficacy of VR exercise protocols or the reliability of VR movement analysis in sport contexts. Future research should explore new VR solutions so that this technology is widely available in sport fields and should investigate the impact of VR on a larger sample of participants. The use of a head-mounted display for an in-field movement analysis of athletes and the early detection of risk factors during games could be useful.

## Figures and Tables

**Table 1 medicina-60-01000-t001:** Search strategy.

***PubMed*** (“lesion” OR “rupture” OR “injury”) AND (“virtual reality” OR “mixed reality” OR “extended reality”) AND (“soccer” OR “football”)
***Scopus*** (“lesion” OR “rupture” OR “injury”) AND (“virtual reality” OR “mixed reality” OR “extended reality”) AND (“soccer” OR “football”)
***Web of Science*** (“lesion” OR “rupture” OR “injury”) AND (“virtual reality” OR “mixed reality” OR “extended reality”) AND (“soccer” OR “football”)

## Data Availability

Data sharing is not applicable to this article.

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
