# Peer review of "The Role of Virtual Reality in the Management of Football Injuries"

_medicina, 2024, doi:10.3390/medicina60061000_

Round 1

Reviewer 1 Report

Comments and Suggestions for Authors

Thank you for the study design. My general suggestion about the study is that I think it would be more appropriate to use the term "soccer" instead of "football" because the word "football" is confused with American football in the literature.

I must say that the introduction is very disorganised and written independently. Although there are many studies written about soccer, a more organised and flowing introduction could have been written.

Although the study is written as a compilation, it contains deficiencies in terms of its purpose and integrity. I think that it would be more appropriate to carry out such studies in the form of a systematic review instead of a compilation and in a way to draw conclusions and suggestions.

The applications and their effects seem to be summarised and presented under headings. Instead, tables can be created and presented in a more organised and understandable way.

Comments on the Quality of English Language

Thank you for the study design. My general suggestion about the study is that I think it would be more appropriate to use the term "soccer" instead of "football" because the word "football" is confused with American football in the literature.

I must say that the introduction is very disorganised and written independently. Although there are many studies written about soccer, a more organised and flowing introduction could have been written.

Although the study is written as a compilation, it contains deficiencies in terms of its purpose and integrity. I think that it would be more appropriate to carry out such studies in the form of a systematic review instead of a compilation and in a way to draw conclusions and suggestions.

The applications and their effects seem to be summarised and presented under headings. Instead, tables can be created and presented in a more organised and understandable way.

Reviewer 2 Report

Comments and Suggestions for Authors

would like to thank the authors for this review.

However, some major points need to be taken into consideration:

  1. The introduction section is a bit chaotic. I suggest that the authors organize this section better with subsections. It is necessary to highlight the primary objective of your review and the need for VR techniques in prevention and rehabilitation. 
  2. I suggest that authors create tables with current studies on athletes, regardless of their sport type, about Vr in Injury Prevention and Vr in Post-Injury Application. 
  3. In the conclusion section, highlight the need for VR techniques in Football and how important it would be for future studies to explore this technique further. 
Comments on the Quality of English Language

I suggest authors be more careful with English grammar. Some minor faults have been detected throughout the text. 

Round 2

Reviewer 1 Report

Comments and Suggestions for Authors

I see that the authors have made some arrangements to improve the study, but I still do not think that this study will be sufficient for the quality of your journal. I think that for the journal, especially review studies should be written with a systematic review and a literature to be analysed in more depth.

Comments on the Quality of English Language

Thank you for the study design. My general suggestion about the study is that I think it would be more appropriate to use the term "soccer" instead of "football" because the word "football" is confused with American football in the literature.

I must say that the introduction is very disorganised and written independently. Although there are many studies written about soccer, a more organised and flowing introduction could have been written.

Although the study is written as a compilation, it contains deficiencies in terms of its purpose and integrity. I think that it would be more appropriate to carry out such studies in the form of a systematic review instead of a compilation and in a way to draw conclusions and suggestions.

The applications and their effects seem to be summarised and presented under headings. Instead, tables can be created and presented in a more organised and understandable way.

Author Response

Thank you for the comment. 

We have made efforts to improve our review.

The heterogeneity of the articles allowed the possibility of writing a narrative review.

Reviewer 2 Report

Comments and Suggestions for Authors

I want to thank the authors for the revised version of their article.

This manuscript can now be published.

Author Response

We would thank the reviewer for the comment.